



# Resolving the influence of temperature forcing through heat conduction on rockglacier dynamics: a numerical modelling approach

Alessandro Cicoira[1], Jan Beutel[2], Jérome Faillettaz[1], Isabelle Gärtner-Roer[1], and Andreas Vieli[1]

[1]Department of Geography, University of Zurich, Zurich, Switzerland
[2]Computer Engineering and Networks Laboratory, ETH, Zurich, Switzerland

*Correspondence to:* Alessandro Cicoira (alessandro.cicoira@geo.uzh.ch)

**Abstract.** In recent years, observations have highlighted seasonal and inter-annual variability in rockglacier flow. Temperature forcing, through heat conduction, has been proposed as one of the key processes to explain these variations in kinematics. However, this mechanism has not yet been quantitatively assessed against real-world data.

We present a 1-D numerical modelling approach that couples heat conduction to an empirically derived creep model for ice-rich frozen soils. We use this model to investigate the effect of thermal heat conduction on seasonal and inter-annual variability in rockglacier flow. We compare the model results with borehole temperature data and surface velocity measurements from the PERMOS and PermaSense monitoring network in the Swiss Alps. We further conduct a model sensitivity analysis in order to resolve the importance of the different model parameters. Using the prescribed empirically derived rheology and observed near-surface temperatures, we are able to model the right order of magnitude of creep flow. However, both inter-annual and seasonal variability are underestimated by an order of magnitude, implying that heat conduction alone can not explain the observed variations. Therefore, non-conductive processes, likely linked to water availability, dominate the short-term velocity signal.

## 1 Introduction

For several rockglaciers worldwide and especially in Switzerland surface displacements have been calculated over long time periods (Chaix, 1923; Wahrhaftig and Cox, 1959; Francou and Reynaud, 1992; Berthling et al., 1998) by using position time series of landmark features (boulders). Since these early investigations, velocity variability has been detected on a multi-yearly scale. In the past decades, starting with some measurements on Gruben rockglacier (Haeberli, 1985), seasonal velocity variability has been observed on such creeping periglacial landforms. Even though differences exist between individual rockglaciers, velocity peak maxima are in general observed between summer and early winter and minima between spring and early summer (Delaloye et al., 2010). In the past years, advances in monitoring techniques and the introduction of continuously measuring D-GPS loggers (Buchli et al., 2012) have confirmed the previous observations on several rockglaciers and have further highlighted velocity peaks on a daily to a weekly scale predominantly present during the melt season (Wirz et al., 2016a; Kenner et al., 2017; Buchli et al., 2018).



In order to explain the above introduced observations, classical concepts from related disciplines - geotechnical engineering and glaciology - have been applied to rockglacier research. Inter-annual velocities have been compared against climatic variables and external temperature forcing has been proposed as one of the key factors controlling the observed long term flow variations (Roer et al., 2005; Krainer and He, 2006; PERMOS, 2016). Similarly, temperature forcing has also been suggested

as one of the most important factors controlling rockglacier flow velocity variability on a seasonal scale (Arenson et al., 2002; Kääb et al., 2007; Delaloye et al., 2010; Wirz et al., 2016b). Wirz et al. (2016b) has identified liquid precipitation, snow melt, air and ground temperature as the main factors controlling rockglacier flow on inter-annual, seasonal, and shorter time scales. Previous studies (Johnson (1978); Barsch (1992); Krainer and He (2006) amongst others) highlighted the influence of water on rockglacier and their dynamics, possibly through positive feedback mechanisms along with rising temperatures and decreasing

effective stresses (Ikeda et al., 2008; Buchli et al., 2018).

Even though great improvements have been achieved in this field, our understanding of the processes governing rockglacier dynamics and their relation with external forcings and controlling factors remains at a qualitative level, and many questions remain unanswered. However, it is clear that in order to understand rockglacier dynamics, the complex thermo-hydro-mechanical behaviour of the ice-rich frozen soil and its coupling with the climate have to be considered. In particular, when aiming at un-

derstanding the influence of temperature forcing on permafrost creep and its relative importance on rockglacier dynamics, we have to consider two aspects. On one hand, the thermal regime of a rockglacier is mainly controlled by heat conduction, driven by external temperature forcing (Vonder Mühll et al., 2003; Haeberli et al., 2006). Nevertheless, in some cases other processes have been observed, like air and water advection through the permafrost matrix (Zenklusen Mutter and Phillips, 2012; Luethi et al., 2017; Pruessner et al., 2018). On the other hand, from glaciological studies it is well known that the rate of deforma-

tion of ice is described by a power law (Nye, 1952; Glen, 1955) and depends on ice viscosity, which in turn depends on ice temperature (Mellor and Testa, 1969; Duval et al., 1983). Therefore, heat conduction forced by external surface temperature variations is expected to influence rockglacier creep. However, investigations that quantitatively couple temperature evolution and rockglacier rheology are rare and remain very limited (Kääb et al., 2007; Müller et al., 2016). The numerical modelling study by Kääb et al. (2007) investigated this process, but has two main limitations: it used a rockglacier rheology that has been

derived for pure ice and more importantly, it applied this rheology to a purely synthetic set up and could not directly compare the results to real-world observations.

In this study, we quantify the relative importance of the conductive thermal influence on flow and extend previous research (Kääb et al., 2007) by applying the most up-to date rheological relation available for rockglacier material (Arenson and Springman, 2005a) to four real world rockglaciers and constrain the modelling with observations from borehole measurements,

kinematics surveys and D-GPS observations available from the Swiss Permafrost Monitoring Network PERMOS (here on named PERMOS) and PermaSense monitoring networks.



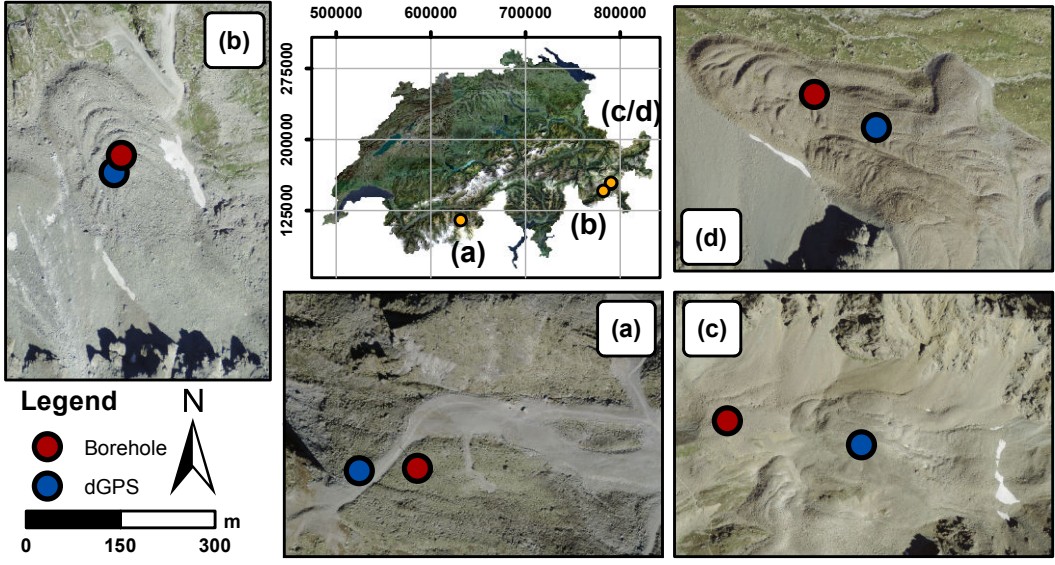

**Figure 1.** Overview of the four case study sites. Overview site map (middle) and aerial view of the four studied rockglaciers (a-d) with locations of boreholes (blue dots) and D-GPS (red dots) according to PERMOS (2016): (a) Ritigraben, (b) Murtèl, (c) Schafberg and (d) Muragl.

## 2 Case Study Sites

For constraining the numerical modelling investigations, we use observational data from four rockglaciers in the Swiss Alps, namely from the rockglaciers Ritigraben located in the Valais, and Murtèl-Corvatsch (for hereon named Murtèl), Schafberg, and Muragl all located in the Engadine (Fig. 1). These rockglaciers have been selected based on the availability of several years of data on highly time resolved surface displacements and subsurface temperatures from boreholes. Further, several borehole deformation profiles are available for different time steps for all rockglaciers. Such type of data is rather unique and made

5  available through the PERMOS monitoring network.

These four rockglaciers cover a wide range of geometric settings and dynamic states: thickness, slope and flow speed from decimetres to several metres per year: for an overview see Table 1. They are located at elevations between 2500 to 2900 m a.s.l. and their aspect is north to north-west. Their lithology mainly consists of crystalline formations, with prevailing granodiorite and schist for Murtèl and gneiss for Muragl, Schafberg and Ritigraben. The internal structure and deformation profiles are

10  known for all four rockglaciers from boreholes. Their motion is dominated by a few meter thick shear horizon at 18 to 30 m depth (Haeberli et al., 1998; Arenson et al., 2002; Lugon and Stoffel, 2010). Laboratory shear experiments have been undertaken on cores from boreholes for the two rockglaciers Murtèl and Muragl and were used by Arenson and Springman (2005a) to derive an empirical creep-rheology (Arenson and Springman, 2005b), which is also used in the flow-modelling investigations of this study (for details see Sect. 3.3).



**Table 1.** Summary of field sites. The borehole locations are given in the CH1903+ coordinate system. The values of the geometrical and physical properties of the rockglaciers are given (thickness, slope, volumetric ice content and thermal diffusivity) as discussed in Sect. 2 Case Study Sites. For the ice content an acceptable range of values is proposed according to the literature. The mean observed bottom temperature (from summer 2006 to summer 2015) and surface velocity (from summer 2009) are reported, as well as the inter-annual and seasonal velocity variations relative to the mean velocity (from summer 2012 to summer 2016).

| Rock glacier | Borehole location | Thickness [$m$] | Surface slope | $w_i$ | $\kappa$ [$m^2 d^{-1}$] | Bottom temp. [°C] | Mean velocity [$ma^{-1}$] | Inter-annual variation | Seasonal variation |
|---|---|---|---|---|---|---|---|---|---|
| Ritigraben | N: 1113751 E: 2631738 | 18 | 27° | $30-70\%$ | 0.18 | $-0.5$ | 1.5 | 25% | 45% |
| Murtèl | N: 1144718 E: 2783160 | 27 | 12° | $70-100\%$ | 0.15 | $-1.2$ | 0.12 | 41% | – |
| Schafberg | N: 1152598 E: 2790943 | 25 | 18° | $30-100\%$ | 0.15 | $-0.1$ | 0.3 | 33% | 39% |
| Muragl | N: 1153722 E: 2791102 | 20 | 20° | $30-70\%$ | 0.18 | $-0.1$ | 1.4 | 25% | 14% |

## 2.1 Ritigraben

The Ritigraben rockglacier is located above the village of Grächen (VS) and origins from the northern slope of the Gabelhorn (3135 m a.s.l.). It develops a simple linear flow lobe of about 500 m lenght on a steep slope (27°in the proximity of the borehole, Fig. 1 and Table 1) and terminates at the upper end of the Ritigraben gully. The surface is disturbed by the ski slope facilities that have been built on the rockglacier. Accounting for the steep slope and the geometrical setting, the flow unit is only 20 meters thick and the flow velocities are rather high. Continuous D-GPS measurements provide velocity data since 2012, showing a mean value of $1.4 \, \mathrm{m\,a^{-1}}$ and strong seasonal and inter-annual variations of more then $45\%$ and $25\%$ respectively, see Table 1. Even though no ice cores have been analysed, ice content has been estimated in previous studies at 30%-70% (Lugon and Stoffel, 2010; Luethi et al., 2017). Borehole measurements since 2002 show warm permafrost temperatures close to the melting point and the progressive development of a Talik at a depth between 10 and 12 meters (Zenklusen Mutter and Phillips, 2012), which has been related to the influence of water infiltration and air circulation (Luethi et al., 2017).

## 2.2 Murtèl-Corvatsch

The Murtèl rockglacier originates from the north-wall of Piz Murtèl (3432 m a.s.l.) and is characterized by a single lobe of 27 m thickness with well developed surface morphology of lobate furrows and ridges that can be attributed to compressive flow, and buckle and folding (Fig. 1a, Frehner et al. (2015)). Consistent with a low surface slope of 12°, this rockglacier is with 0.14



15 m a$^{-1}$ rather slowly flowing (Müller et al. (2014); see Table 1). Murtèl is probably the best studied rockglacier in the world with continuous temperature monitoring data from boreholes available since 1987 (Haeberli et al., 1988, 1998). The drillings from 1987, 2000 and 2015 (Haeberli et al., 1988; Vonder Mühll et al., 2003) and geophysical investigations (Haeberli et al., 1998; Arenson et al., 2002) revealed relatively ice-rich material in the main rockglacier body with an estimated ice content close to 100%. The temperatures within the main body of the rockglacier are between $-4°$ and $-1°$ Celsius and therefore relatively cold

compared to other instrumented rockglaciers in Switzerland (Vonder Mühll et al., 2003; PERMOS, 2016). Annual velocities are measured since 2009 by geodetic survey of 11 surface markers around the borehole (PERMOS, 2016). Further, several years of borehole deformation data (at time intervals of several months) are available from 1987 to 1994 (Haeberli et al., 1998). The time averaged yearly velocities show an increasing trend, coherent with observations for other rockglaciers throughout the Swiss Alps, as showed in the PERMOS Glaciological Report No. 12-15 (PERMOS, 2016).

## 2.3 Schafberg

The Schafberg rockglacier origins in a cirque south of the Piz Muragl ridge, has an extent of less than 300 meters and a surface slope of 18 °(Table 1, Fig. 1c). In the lower part, the rockglacier splits into two separate tongues as a result of a bedrock outcrop. This study focuses on the north-western lobe where in 1997 a borehole has been drilled and temperatures are monitored thereafter in the framework of PERMOS. This lobe has a thickness of approximately 26 m and an flow speed of $0.3\,\mathrm{m\,a^{-1}}$.

Investigations by Vonder Mühll (1993) show a volumetric ice content ranging between from 35% to 100%. Continuous daily velocities are measured by D-GPS within the PermaSense framework since 2012 approximately 200 m upstream of the borehole location and show clear seasonal variations with an amplitude of up to 39% relative to the mean velocity and a rising inter-annual trend (33% increase in the observed period).

## 2.4 Muragl

The Muragl rockglacier is located on the west side of the ridge of Piz Muragl (3156 m a.s.l.) and consists of several generations of overlapping flow units of variable flow speeds (Fig. 1d). The main lobe, where the borehole is located, moves at $1.5\,\mathrm{m\,a^{-1}}$, is approximately 25 m thick and has a surface slope of $20°$ (Table 1). The annual surface motion is available from terrestrial survey since 2009 whereas continuous daily velocities from D-GPS-measurements at the lower end of the lobe are measured since 2012 and indicate clear inter annual (25%) and seasonal variations (14%) (PERMOS, 2016). Ice content has been estimated

from boreholes investigations at $40 - 70\%$ and is found to be very heterogeneous (Arenson et al., 2002). The temperatures within the rockglacier, measured since the drilling in 1999, range from $-3°$ to $0°$ Celsius and a relatively close to the melting point (Vonder Mühll et al., 2003). As for the Murtèl rockglacier and consistent with other observations in the Alps, there is a rising trend of multi-annual velocities (PERMOS, 2016).





## 3   Data and Methods

We designed a suite of 1-dimensional numerical models, based on finite-differences, to simulate the response of viscous and plastic flow to external near-surface temperature forcing. The modelling framework couples heat conduction, forced by external temperature, to a power-law creep relation for ice-rich frozen soils proposed by Arenson and Springman (2005a). The model inputs are the surface slope, the thickness and other physical properties (density, ice content and thermal diffusivity) of the creeping rockglacier, which are all assumed to be homogeneous in time and space. The model is forced by permafrost temperature time-series below the active layer as measured in boreholes. At the lower boundary of the rockglacier a constant temperature value representative of the observed bottom temperature is prescribed. The model is applied to the four real-world cases described in Sect. 2 and the results are compared to observed borehole temperatures and surface flow velocities.

### 3.1   Data overview

Here, we provide a detailed description of the data used for model input and for comparison with the model results.

The surface slope values are calculated on the basis of the swisstopo digital cartography (Federal Office of Topography swisstopo) over a 200 meter long profile along a flow line centred at the D-GPS location and are representative of the average slope of the landform close to the D-GPS positions. For all four study cases vertical profiles of borehole deformation are available at several time steps over few years (Arenson et al., 2002; Kenner et al., 2017). The rockglacier thickness is defined based on these profiles as the vertical distance from the surface to the lower end of the shear horizon. Ice content values between $30\%$ and $100\%$ have been noted in the literature and are mostly based on borehole drillings (Vonder Mühll, 1993; Arenson et al., 2002; Lugon and Stoffel, 2010). The densities of the rockglaciers are calculated as a weighted average between the density values of pure ice $\rho_{ice}$ and solid rock $\rho_{rock}$ from the volumetric ice content $w_i$. The thermal diffusivity $\kappa$ is calculated similarly based on the thermal diffusivity values of pure ice ($0.1$ m$^2$ day$^{-1}$) and quarz ($0.35$ m$^2$ day$^{-1}$) as proposed by Williams and Smith (1989). Borehole temperature time series are available from the PERMOS database for the four study cases from Spring 2002 to Autumn 2015 with a time resolution of six hours (the time series of Muragl terminates in Autumn 2014). For our modelling, we re-sample these data to a daily average. For all four study cases, some data gaps occur due to sensor failure, in particular at depth. For temperatures at depth just below the active layer, which are used as forcing input for the model, the data gaps are below a few months apart from the case of Muragl, for which temperature data in the period comprised between August 2008 and April 2009 are missing. Missing temperature data are linearly interpolated. Note that this linear interpolation does not aim to reconstruct exact real temperatures, but rather bridges the gaps in order to create a continuous temperature time series that can be used as model input. This means the modelled short-term velocity variations should not be analysed near these gap filled periods. Figure 2 shows the measured temperatures for all case study sites.

Several type of velocities data are available. For Murtèl and Muragl rockglacier mean annual surface velocities are available between 2009 to 2015 from terrestrial surveys with total station from the PERMOS network. For all study cases but Murtèl, daily velocities from continuous single frequency D-GPS measurements are available since 2012 from the PermaSense network. For a summary of the study case sites see Table 1.





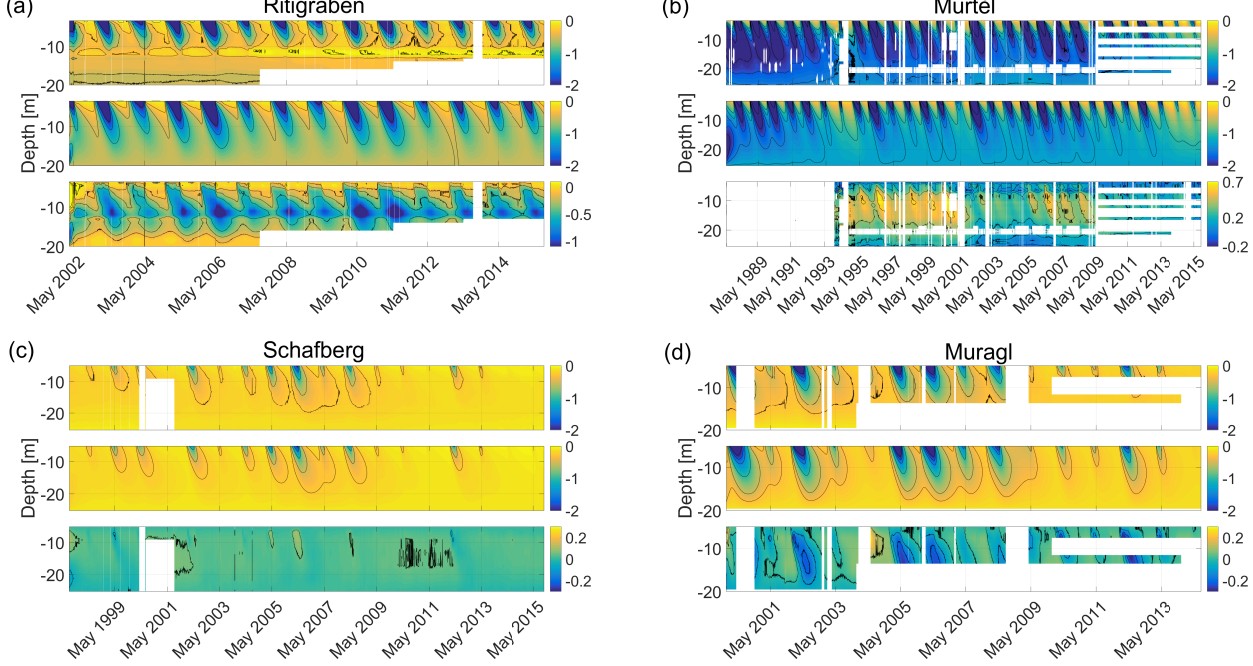

**Figure 2.** Contour plots of ground temperatures time series (color coded) for (a) Ritigraben, (b) Murtèl, (c) Schafberg and (d) Muragl rockglacier. Each panel shows, from top to bottom, measured temperatures, modelled temperatures and differences between the measured and the modelled temperatures. Note the different temperature scales of the lower sub panels with the temperature differences.

## 3.2 Heat conduction model

We model vertical heat conduction throughout the rockglacier unit by solving the diffusion equation for temperature evolution with depth:

$$\frac{\partial T}{\partial t} = \kappa \frac{\partial^2 T}{\partial z^2}, \tag{1}$$

5 where $T$ is the permafrost temperature, $z$ the vertical coordinate, $t$ the time and $\kappa$ the thermal diffusivity of the rockglacier material. At the upper boundary the observed temperature history just below the active layer depth is prescribed. At the bottom of the rockglacier (below the shear horizon) a constant temperature value corresponding to the time average of the observations is prescribed. The initial condition is prescribed from the measured vertical temperature profile at the first time step of the simulation. The temporal resolution of the model is 1 day, its vertical resolution is 0.1 meters. Convective and advective heat

10 fluxes and any influence from basal heating due to frictional processes, heat dissipation from deformation and geothermal heat flux are not considered in this model.





### 3.3 Ice-creep model

For modelling ice creep we use the empirically derived creep-relation proposed by Arenson and Springman (2005a). The samples used to derive this relation have been cored from Murtèl and Muragl rockglacier, also investigated in this study, and are described in detail in Arenson and Springman (2005b). The creep relation is a modified Glen's flow law, that relates strain rate $\dot{\varepsilon}$ to a stress invariant $\sigma_e$ as proposed by Von Mises (1913), taking into account the volumetric ice content $w_i$ and the temperature $T$ of rockglacier material:

$$\dot{\varepsilon} = A(T, w_i)\sigma_e^{n(w_i)}. \tag{2}$$

The flow law exponent $n$ linearly depends on ice content only:

$$n = 3w_i \tag{3}$$

and the creep parameter $A$ depends on temperature and ice content by:

$$log A = \frac{2}{1+T} + b(w_i), \tag{4}$$

where $b(w_i)$ is a function of the ice content:

$$b = log(5 \times 10^{-11} e^{-10.2w_i}). \tag{5}$$

Assuming an infinitely wide surface parallel slab, the shear stress $\sigma_e$ at a depth $z$ is given by

$$\sigma_e = \frac{1}{\sqrt{3}}\rho_r gz \frac{\partial s}{\partial x}, \tag{6}$$

where $\rho_r$ to the density of the rockglacier material, $g$ the constant of gravity and $\frac{\partial s}{\partial x}$ the slope of the ice surface. Note that the density $\rho_r$ also depends on the ice density by:

$$\rho_r = \rho_s(1 - w_i) + \rho_i w_i, \tag{7}$$

where $\rho_i$ and $\rho_s$ are the densities of ice and sediment particles respectively. As in our case the ice thickness of the rockglacier landform is fixed, any variation in ice content $w_i$ will change the density and as consequence the shear stress $\sigma_e$ (see Eq. (6)).

Given the temperature field and assuming an inclined infinite parallel slab, the velocities are solved from Eq. (2) through vertical integration under the assumption of zero displacement at the lower boundary. The physical properties of the material are considered constant in time and homogeneous in space. The temporal and the spacial model resolution are the same as for the heat-conduction model. Despite its limitations, the proposed constitutive relation is one of the most up-to date on rockglacier material and has the great advantage of being based on laboratory experiments.

### 3.3.1 Calibration of the model

The model is calibrated to best fit the average observed surface velocities by varying the ice content parameter within the range of literature values. Because of the mathematical formulation of the applied rheology, calibrating the model by varying




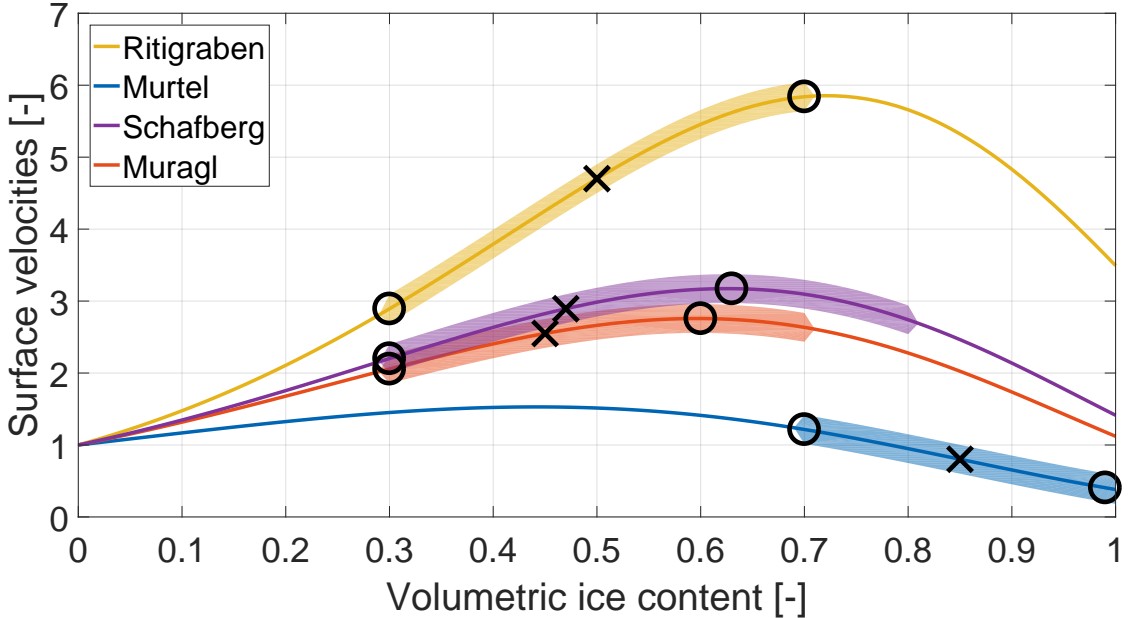

**Figure 3.** Normalized surface velocities for the four case studies with volumetric ice content values. For each rockglacier the velocities are normalized with the velocity value corresponding to $0\%$ ice content. The coloured buffer around the curve shows the range proposed by the literature. The black circles show for each curve the maximum and the minimum possible velocity values within this range, also representing the uncertainty range for the modelling. The black cross shows the chosen ice content value for the modelling, i.e. the mid value of the latter range.

30  the volumetric ice content gives a concave curve for the mean velocities, with a maximum corresponding to around $60\%$ ice content (see Fig. 3). On one hand, an increasing ice content causes the creep relation exponent to grow resulting in higher surface velocities. On the other hand, a higher volumetric ice content implies a lower density, and thus lower shear stress and deformation. This mathematical artefact is because the thickness of the rockglacier in the modelling is fixed. In reality, for varying ice content values, the rockglacier thickness and velocity would adjust until the shear stress at the base of the rockglacier reaches a critical value.

5  Within the range of possible ice content values proposed in the literature, we further refine the range for which the maximum and the minimum velocities are obtained (black circles in Fig. 3). Here on, we set the ice content parameter to the mean of this range and further assess the effect of the uncertainty range on the velocity results.

We consider further uncertainties in input parameters: slope $\pm 2°$, ice content (corresponding to the maximum and minimum velocity value) and thermal diffusivity $\pm 0.02$ m$^2$ day$^{-1}$.

The creep model strongly depends on the temperature input. In order to assess uncertainties related to the heat conduction model and to take into account all possible heat transfer process, we perform additional numerical experiments forcing the ice-creep relation directly with the observed temperature fields with depth.





### 3.3.2 Consideration of shear horizon

The above described approach for creep does not consider enhanced deformation in the shear horizon, where most of the displacement takes place. In order to investigate the sensitivity of the model to such phenomenon, we perform additional numerical experiments. We approximate the behaviour of the shear horizon with a pseudo-plastic creep relation, by increasing the flow law exponent of Eq. (2) by a factor 4 ($n_{plastic} = 12 * w_i$), similarly to Frehner et al. (2015). The creep parameter A has been reduced by a factor $f_A$ to match the time averaged surface velocities modelled before:

$$\dot{\varepsilon}_{plastic} = f_A(\dot{\varepsilon})A(T, w_i)\sigma_e^{n_{plastic}(w_i)} \tag{8}$$

In this way, we approximate the plastic behaviour of the lower layer to better represent the whole deformation profile of the studied rockglaciers.

### 3.4 Sensitivity experiments

We perform additional synthetic sensitivity experiments in order to explore the influence of the different input parameters on our model results. For these experiments we simulate seasonal temperature forcing by prescribing the temperature below the active layer as sinusoidal function with a mean of $0$ °C, that truncates positive temperatures in order to take into account the zero curtain effect. The initial vertical temperature profile is set to $0$ °C. The model runs for $28$ full annual cycles after which it converged to a quasi steady-state periodic solution. We then analyse the results of the last two successive cycles. We study the sensitivity to different varying seasonal temperature amplitude (corresponding to the minimum winter temperature), temperature at the lower boundary, rockglacier thickness, ice content and thermal diffusivity. We set up a reference scenario with typical values for rockglaciers taken as *Scn1.0* in Table 3. Starting from the reference scenario, we perform $9$ experiments for each parameter, in which we vary the value of the considered parameter by a factor from $0.2$ up to $1.8$, with the other parameters being fixed. The numbers in the experiment name in Tab 3 refer to the multiplication factor of the parameters relative to the reference scenario. Only for the experiments on the volumetric ice content parameter the multiplications factors are different, as shown in Table 3.

Additionally, the thickness sensitivity experiments have been repeated using the pseudo-plastic rheology (Eq. (8)) in order to investigate the effect of the presence of the shear horizon in our experiment. In the case of both sets of varying thickness experiments (with and without shear horizon), the bottom temperature for the scenarios with thicknesses less then the one of the reference are always prescribed at $20$ meters depth. For these shallow depths, prescribing a constant temperature would unrealistically constrain the temperature field. For all the values and the results of this analysis we refer to Table 3.



## 4   Results

### 4.1   Modelled temperatures

The modelled and measured temperatures are shown in Fig. 2. For Schafberg and Muragl rockglaciers (Fig. 2c and Fig. 2d respectively) the modelled temperatures agree very well with the measurements (temperature differences are below $0.2°$C). For

the case of Murtèl rockglacier, given the prescribed temperatures below the active layer the modelled temperature evolution with depth agrees well with regard to seasonal amplitude and phase with depth. However, between a depth of 5 and 20 meters, temperatures are in particular during cold seasonal phases slightly underestimated, but the differences stay below $0.5°$C. For Ritigraben rockglacier, as shown in Fig. 2a, in a depth between 8 and 12 meters and in particular in early summer, observed temperatures are substantially higher (up to $1°$C), which is related to the Talik observed and discussed in Luethi et al. (2017).

### 4.2   Modelled velocities

The observed and modelled surface flow velocities with time are shown in Fig. 4 for using the modelled temperatures (solid blue line), for using the observed temperature fields (red solid line), and for using the pseudo-plastic rheology (yellow solid line) with modelled temperatures. The resulting maximum and minimum velocities accounting for uncertainties in the input parameter of ice content, slope and thermal diffusivity are shown with two black dashed lines.

For the chosen ice content values within the proposed range, we obtain the right order of magnitude of the observed surface velocities for all four case study rockglaciers. For Ritigraben and Muragl rockglaciers the modelled velocities are smaller (by a factor 2), for Murtèl the average velocity match and for Schafberg the modelled velocities are overestimated (by a factor 3) in comparison to the observed ones.

The modelled amplitudes in seasonal and multi-annual velocity variations (values in Table 2, solid blue, yellow, and red

line in Fig.4) are in general rather low: below $10\%$ relative to the mean speed. In comparison, the observed variations in flow (solid green and purple lines in Fig. 4, values in Table 1) are one order of magnitude higher. This result does not change when considering uncertainties in the input parameters (dashed black lines in the same figure).

For the three rockglaciers with continuous *D-GPS* measurements, we are also able to compare the phase of the seasonal variations. The modelled velocity maxima occur in late winter and are substantially delayed in comparison to the observed velocity peaks in autumn. The above findings (amplitude underestimation and phase shift), do not change when the observed temperature fields are used as input for the creep model. The exception is the case of Ritigraben, where, as discussed above,

5   substantial discrepancies between velocities obtained using modelled (blue solid line) and observed (red solid line) temperatures occur. When using the observed temperature field, the modelled and the observed seasonal velocity variations are phase synchronous (see red and green line in Fig. 4).

For all rockglaciers the discrepancies found between observed and modelled velocity variations do not improve when using the pseudo-plastic creep model. In the contrary, the seasonal velocity amplitude further reduces and the phase shift increases

10   further (see yellow line in Fig. 4).





**Figure 4.** Observed and modelled surface flow velocities for (a) Ritigraben, (b) Murtèl, (c) Schafberg and (d) Muragl rockglaciers. The upper sub-figure shows the observed subsurface temperature used as model input. The lower sub-figure shows the modelled and observed velocities. The modelled velocities are shown for using the modelled temperatures (solid blue line), the observed temperature fields (red solid line), and the pseudo-plastic rheology (yellow solid line). The uncertainty range resulting from variations in slope ($\pm 2°$), ice content (within the proposed range, see Fig. 3 and Table 2) and thermal diffusivity ($\pm 0.02 \, \mathrm{m^2 \, day^{-1}}$) is plotted with a black dashed line. The modelled velocities are compared with the observed velocities from terrestrial surveys (dark solid red line with black dots) and with D-GPS measurements (green solid line).



**Table 2.** Summary of modelling inputs and results. Values of geometrical and physical input parameters for the modelling are listed (thickness, slope, volumetric ice content and thermal diffusivity). The mean modelled surface velocity and its relative inter-annual and seasonal variations are reported in the last three columns.

| Rock glacier | Thickness [$m$] | Temp. input depth [$m$] | Slope | $w_i$ | $\kappa$ [$m^2 d^{-1}$] | Mean surface velocity [$ma^{-1}$] | Inter-annual variation | Seasonal variation |
|---|---|---|---|---|---|---|---|---|
| Ritigraben | 18 | 3.5 | 27°±2° | $50 \pm 20\%$ | $0.18 \pm 0.02$ | 0.59 | 2% | 8% |
| Murtèl | 27 | 3.5 | 12°±2° | $85 \pm 15\%$ | $0.15 \pm 0.02$ | 0.12 | 5% | 2% |
| Schafberg | 25 | 5.2 | 16°±2° | $48 \pm 18\%$ | $0.15 \pm 0.02$ | 0.69 | 4% | 5% |
| Muragl | 20 | 4.5 | 20°±2° | $45 \pm 15\%$ | $0.18 \pm 0.02$ | 0.46 | 6% | 5% |

## 4.3 Sensitivity experiments

The results of the sensitivity experiments are plotted in Fig. 5 and Fig. 6 and are summarized in Table 3.

In Fig. 5, the time averaged velocity values for the different experiments and scenarios are shown normalized to the mean values of the reference scenario. The left panel shows the results of the experiments investigating the thermal regime of the rockglacier for varying sub-surface temperature forcing amplitude, bottom and initial temperature, and thermal diffusivity. The initial condition experiment shows no differences (purple line constant at 1) and demonstrates that the model converged to a quasi steady-state solution after the 28 annual cycles. A reduction in bottom temperature to $-0.2$ °C leads to an increase in the mean velocity of $50\%$ and a decrease in seasonal amplitude by a factor 0.2 leads to an increase in the mean velocity of $20\%$ (Fig. 5). Increasing the thermal diffusivity leads to only very slightly increased mean velocity values (less then $1\%$). The modelled flow is more sensitive to the varying geometrical and physical parameters (thickness, slope and volumetric ice content; right panel in Fig. 5). The velocities strongly vary with thickness and slope, in accordance with the governing equations (Eq. (2) and (6)), following a power law with variations of almost $400\%$ and $600\%$ respectively. For varying ice content the mean velocities show variations of up to $70\%$ for the used parameter range (see Sect. 3.3).

In Fig. 6, the velocities of the different sensitivity experiments normalized with their mean are presented and a summary is given in Table 3. The velocity seasonal response to different amplitudes in surface winter temperature forcing is small and stays below $7\%$ even for an $80\%$ increase in the temperature amplitude. The sensitivity of the velocity variations to varying bottom temperature are slightly higher, but remain below $9\%$ in the given range. For the thickness experiment, a velocity variation of up to $80\%$ is obtained when considering a $4\,m$ thick rockglacier, representing an extreme and unrealistic scenario. For a more realistic lower bound of rockglacier thickness of 16 m the seasonal velocity variations stay below $10\%$. For thickness variations with the pseudo-plastic creep relation the sensitivity is even smaller. Variations in volumetric ice content and thermal diffusivity lead to slightly increased seasonal velocity variability of $12\%$ and $16\%$, respectively. Thus in general, within a reasonable range of input parameters, the modelled seasonal variations in surface velocity remain more than one order of magnitude below the observed seasonal variations.




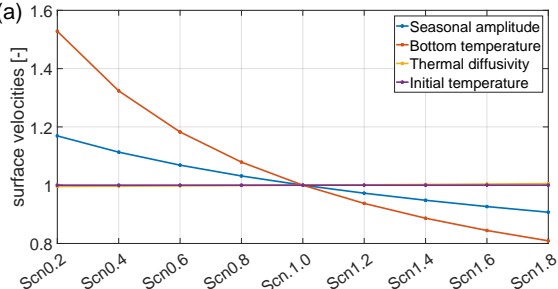
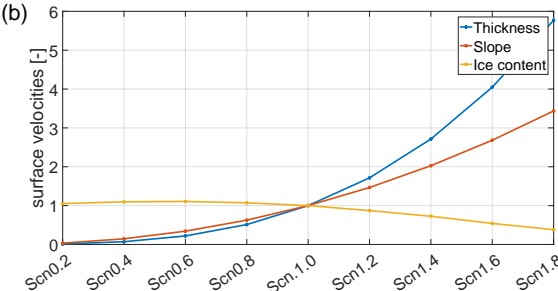

**Figure 5.** Plots of the time averaged values of surface velocities for the sensitivity experiments with the different parameter scenarios. The mean velocities are normalized with the mean values of the reference scenario *Scn1.0*. Panel (a) shows the results for varying surface temperature forcing amplitudes (blue line), bottom temperature values (red line), thermal diffusivity parameters (yellow line) and initial condition temperature values (purple line). The yellow line is hidden behind the purple line. Panel (b) shows the results for varying thickness values (blue line), slope angles (red line), and ice content values (yellow line).

**Table 3.** Parameters and results of the sensitivity experiments. The first column indicates the scenario with a label in the form of *Scnα*, where $\alpha$ refers to the value of the multiplication factor for the investigated parameter relative to the reference values. Each column represents a set of experiments for one variable parameter, with the first number in each column referring to the input parameter value. The numbers in the bracket of each column refer to seasonal variations as a percentage of the mean value on the left side and the phase shift relative to the reference scenario on the right side (with a cycle of $2\pi$ referring to one year).

| | Seasonal amplitude [°C] | Bottom temperature [°C] | Thickness [m] | Shear horizon depth [m] | Ice content [%] | $\kappa$ [$m^2 d^{-1}$] |
|---|---|---|---|---|---|---|
| Scn0.2 | $-0.8\ [3.1\% - \frac{\pi}{14}]$ | $-0.2\ [8.4\% + \frac{\pi}{5}]$ | $4\ [80.4\% - \frac{\pi}{8}]$ | $4\ [68.8\% - \frac{2\pi}{3}]$ | $40\ [11.6\% - \frac{\pi}{8}]$ | $0.03\ [0.8\% - \frac{\pi}{8}]$ |
| Scn0.4 | $-1.6\ [4.5\% - \frac{\pi}{17}]$ | $-0.4\ [7.6\% + \frac{\pi}{7}]$ | $8\ [43.6\% - \frac{\pi}{11}]$ | $8\ [30.8\% - \frac{\pi}{2}]$ | $48\ [9.5\% - \frac{\pi}{8}]$ | $0.06\ [1.4\% - \frac{\pi}{8}]$ |
| Scn0.6 | $-2.4\ [5.3\% - \frac{\pi}{23}]$ | $-0.6\ [6.9\% + \frac{\pi}{11}]$ | $12\ [21.0\% - \frac{\pi}{25}]$ | $12\ [13.2\% - \frac{\pi}{4}]$ | $55\ [8.0\% - \frac{\pi}{8}]$ | $0.09\ [2.2\% - \frac{\pi}{12}]$ |
| Scn0.8 | $-3.6\ [5.7\% - \frac{\pi}{37}]$ | $-0.8\ [6.4\% + \frac{\pi}{23}]$ | $16\ [9.8\% + \frac{\pi}{20}]$ | $16\ [5.8\% - \frac{\pi}{15}]$ | $63\ [6.7\% - \frac{\pi}{9}]$ | $0.12\ [3.9\% - \frac{\pi}{90}]$ |
| **Scn1.0** | $\mathbf{-4.0}\ [6.1\% + 0]$ | $\mathbf{-1.0}\ [6.1\% + 0]$ | $\mathbf{20}\ [6.1\% + 0]$ | $\mathbf{20}\ [2.0\% + 0]$ | $\mathbf{70}\ [6.1\% + 0]$ | $\mathbf{0.15}\ [6.1\% + 0]$ |
| Scn1.2 | $-4.8\ [6.3\% + \frac{\pi}{37}]$ | $-1.2\ [5.8\% - \frac{\pi}{26}]$ | $24\ [2.8\% + 0]$ | $24\ [0.9\% + \frac{\pi}{4}]$ | $78\ [5.6\% + \frac{\pi}{9}]$ | $0.18\ [8.5\% - \frac{\pi}{60}]$ |
| Scn1.4 | $-5.6\ [6.5\% + \frac{\pi}{17}]$ | $-1.4\ [5.5\% - \frac{\pi}{14}]$ | $28\ [1.6\% - \frac{\pi}{13}]$ | $28\ [0.4\% + \frac{\pi}{2}]$ | $85\ [5.3\% + \frac{\pi}{5}]$ | $0.21\ [11.0\% - \frac{\pi}{30}]$ |
| Scn1.6 | $-6.4\ [6.6\% + \frac{\pi}{11}]$ | $-1.6\ [5.4\% - \frac{\pi}{11}]$ | $32\ [1.2\% - \frac{\pi}{9}]$ | $32\ [0.2\% + \frac{3\pi}{4}]$ | $93\ [5.0\% + \frac{\pi}{4}]$ | $0.24\ [13.5\% - \frac{\pi}{20}]$ |
| Scn1.8 | $-7.2\ [6.8\% + \frac{\pi}{9}]$ | $-1.8\ [5.2\% - \frac{\pi}{10}]$ | $36\ [0.9\% - \frac{\pi}{9}]$ | $36\ [0.1\% + \pi]$ | $100\ [4.8\% + \frac{\pi}{3}]$ | $0.27\ [15.9\% - \frac{\pi}{15}]$ |





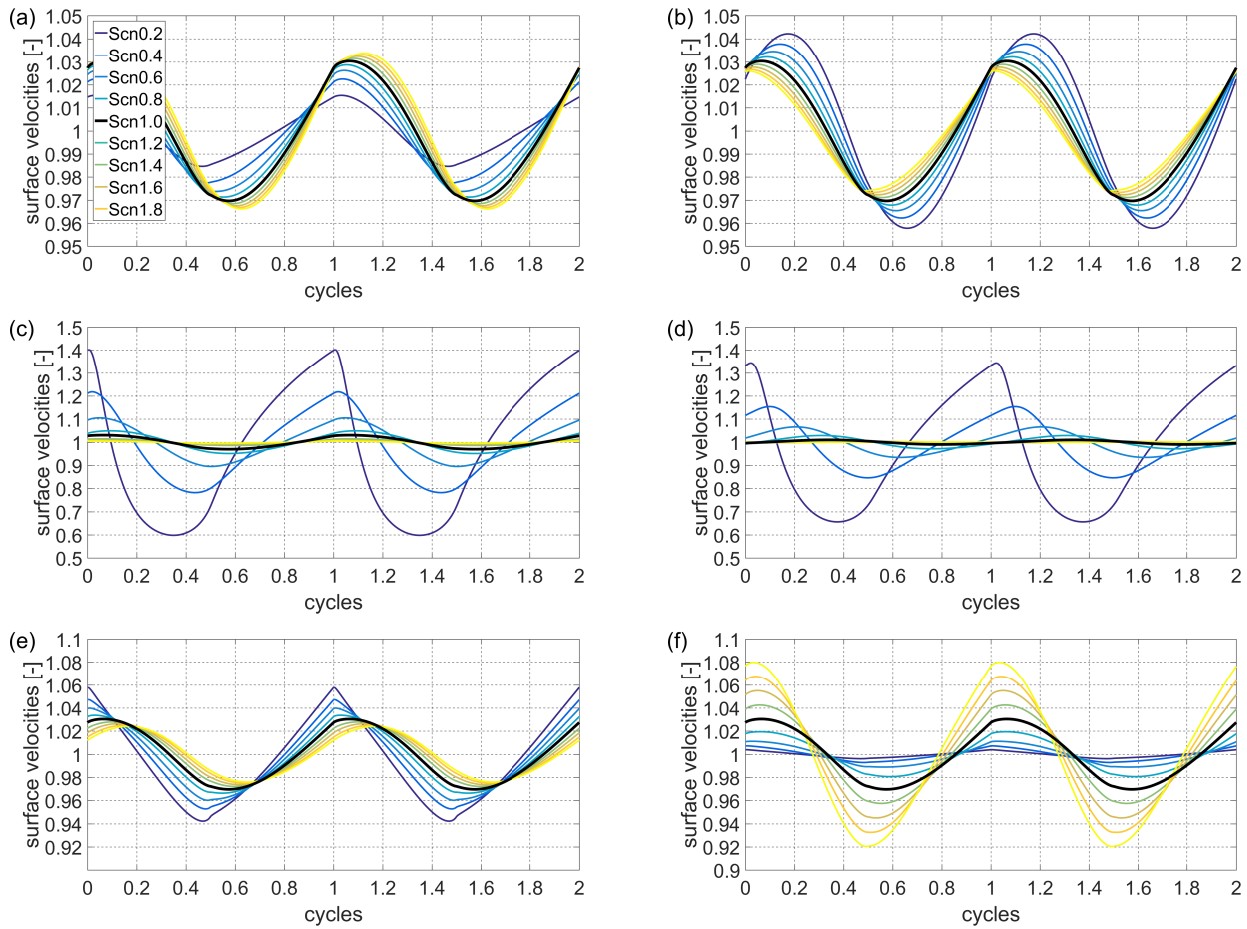

**Figure 6.** Results of the modelling sensitivity experiments of the model to (a) seasonal amplitude (winter minimum temperature), (b) bottom temperature, (c) thickness, (d) thickness (with shear horizon), (e) volumetric ice content and (f) thermal diffusivity. The surface velocities normalized with their mean are plotted for each experiment with time, given here as the number of cycles (after the 28th cycle). One cycle corresponds to one year. For all experiments the different scenarios are color coded as illustrated in the legend of panel (a); the reference scenario is plotted with a thicker solid black line.

## 5  Discussion

In this study, we developed a simple model approach to investigate the dynamical behaviour of rockglaciers with the aim of resolving the influence of external temperature forcing through heat conduction on rockglacier surface velocities. When choosing volumetric ice contents within a physical range of values proposed in the literature (see Sect. 2 and Sect. 3.3), for

5  all the case studies the right order of magnitude of measured mean surface velocities is obtained. However, inter-annual and

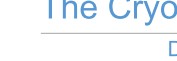

seasonal variations are for all four cases strongly underestimated, being at least one order of magnitude smaller than the observed ones. In the following sections the results are d

## 5.1 Temperatures modelling

We model rockglacier temperature evolution based on near surface temperatures as measured below the active layer (see
sect. 3.2). In some cases data gaps are present and linear interpolation of the data is used. The data gaps are short (below a few months) and don't affect the overall modelling, but interpretation of the modelled velocities for these periods has to consider this issue.

The assumption of constant bottom temperature agrees well with the observed borehole temperatures. This is further supported by the good agreement between the modelled velocities from prescribed observed and modelled temperatures. We
assume the physical properties of the rockglacier (density, ice content and thermal diffusivity) to be constant in time and homogeneous in space, which seems justified at the considered short (seasonal to multi-annual) time scales and supported by the good performance of the temperature evolution model.

For Schafberg (Fig. 2c) and Muragl (Fig. 2d) rockglaciers, we can very well reproduce the observed temperature fields. For Ritigraben and Murtèl (Fig. 2b and Fig. 2d) our results show some disagreement with seasonal pattern, in particular at $12-15$
m depth. At Ritigraben, this disagreement can be explained by the influence of a Talik caused by air and water advection (Luethi et al., 2017), which refers to processes that are not included in our modelling. For Murtèl, the cause of the discrepancy between modelled and observed temperatures is not clear. A possible explanation of this effect could be related to advective water fluxes or varying thermal conductivity within the rockglacier body, likely linked to the variable water content. However, the results of our modelling for the four study cases, in combination with the available borehole temperature observations,
allow us to confidently model and analyse rockglacier velocities.

## 5.2 Ice-creep modelling

Using the modelled and observed temperature fields respectively, we force the empirical creep relation for rockglacier material. Additionally, we run a separate experiment with the pseudo-plastic rheology to investigate the impact on the model from including enhanced deformation within the shear horizon.

### 5.2.1 Absolute velocities

When applying the creep-rheology of Arenson and Springman (2005a) and using acceptable and uniform values of the model input parameters, we obtain the correct order of magnitude of the average observed surface velocities for all case studies.

For Murtèl the modelled mean surface velocities match the observations. For Ritigraben and Muragl the modelled mean velocities are between $30\%$ and $40\%$ of the observed ones. This result is consistent with the observed borehole deformation
contribution from above the shear horizon, accounting for $10\%$ to $30\%$ of the total deformation (Arenson et al., 2002). This finding suggests that the rheology proposed by Arenson and Springman (2005a) may not be applicable to describe the rheology





of the shear horizon of a rockglacier. On the contrary, for Schafberg the modelled velocities overestimate the observations. This mismatch can likely be explained by using a too high input thickness resulting from the distance between D-GPS and borehole location. The rockglacier thickness at the location of the observed velocities by D-GPS is not known and the used thickness has been taken from the borehole on the lobe further down which is less steep. The observed flow magnitude can be matched

almost perfectly when using a thickness of $17\,m$, which has observed in a nearby borehole at the same field site (Arenson et al., 2002), and which can be expected for a steeper surface.

### 5.2.2    Seasonal and multi-annual variations

For all rockglaciers, we find that both seasonal and in particular inter-annual variations are strongly underestimated. This result is coherent also when considering relative velocity variations (see Fig. 4). Especially the results for Murtèl and Schafberg

rockglacier show with $3\%$ to $4\%$ very small seasonal variations. This result is not due to an underestimation in the seasonal temperature variations, as confirmed by the comparison between modelled and measured temperatures and further corroborated by the results of our modelling constrained with the observed temperatures. In fact, the latter forcing indirectly takes into account all non-conductive processes governing the temperature field. The lower sensitivity of rockglacier Murtèl and Schafberg compared to the other two can be explained by their higher thickness. For thicker rockglaciers temperature variations at the

surface reach in relative terms shallower depth and hence do not affect the near bottom layers where most of the deformation occurs. For the thinner Muragl and Ritigraben, the modelled seasonal variations are with $5\%$ and $8\%$ substantially higher, but still a factor 3 to 5 below the observations.

In general, our modelled seasonal variations for the four rockglaciers as well as for the sensitivity experiments are consistent with the obtained $3\%$ to $11\%$ by the earlier idealised modelling study of Kääb et al. (2007). The higher variations for Riti-

graben and Muragl are likely a result of the higher temperature-sensitivity of the rheology by Arenson and Springman (2005a) compared to the rheology based on Glen used in Kääb et al. (2007) for the case of warm permafrost (Müller et al., 2016).

Consistent with the results for similar thicknesses of Kääb et al. (2007), we also find that including a shear horizon in the modelling (by using the pseudo-plastic rheology) decreases the sensitivity of the seasonal variations in flow to temperature forcing. This result further corroborates the underestimation of seasonal and inter-annual variations in our modelling compared

to the observations. It is unlikely that this underestimation is a result of an insufficient sensitivity of the used rheology of Arenson and Springman (2005a) to temperature. In fact, this rheology is based on laboratory deformation-experiments on core-material from real rockglaciers. Unfrozen water is known to have a significant influence on frozen soils creep (Arenson et al., 2006; Moore, 2014), but the influence of interstitial water on creep is already implicitly taken into account in the adopted empirical creep relation through the dependency on temperature. Further, the discrepancy in the phase shift between modelled

and observed velocity variations would not improve for a more temperature sensitive rheology.

### 5.2.3    Phase shift

A phase lag of about 2-3 months between seasonal summer peak in the observed ground surface temperatures and measured surface velocity has been detected on several rockglaciers including Ritigraben, Schafberg and Muragl (see Fig. 7) and has





partly been attributed to the time it takes for the seasonal temperature signal to propagate into the rockglacier (Kääb et al., 2007; Delaloye et al., 2010; Wirz et al., 2016b). In our modelling this delay is however almost doubled, with the seasonal peak in speed obtained in early January rather than early October. For the pseudo-plastic rheology this delay is further extended by several months.

In contrast, the seasonal winter minima in measured temperatures below the active layer (used as model-input forcing) have only a lag of 2 month on the surface temperatures and seem in phase with the observed velocity minima (Fig. 7). Due to the zero curtain effect there is no clear summer peak in the observed and prescribed near surface temperatures (Fig. 7) and the quantification of the summer peak phase shift therefore ambiguous.

Despite the highly asymmetric seasonal temperature pattern, the resulting modelled surface flow variations are almost symmetric (Fig. 4), which is further supported by the sensitivity experiments using a capped sinusoidal forcing function pattern (Fig. 6). This transformation of the seasonal pattern is on the one hand a result of the diffusion of the temperature signal and on the other hand a result of an integrated contribution of deformation over the entire depth. The seasonal pattern in surface speed variation is therefore neither a direct reflection of the temperature signal at a single depth nor of the depth averaged temperature signal. As a consequence, estimating the phase lag between seasonal variations in surface temperature and surface flow from heat conduction is non-trivial and interpreting phase lags potentially misleading.

The clear overestimation of the time lag in the modelled surface variations is a further sign that the process of heat conduction alone can not explain the observed variations. Infiltration of surface melt water into the permafrost in the summer season could reduce this time lag and through advection of water affect the flow in two ways. Firstly, the infiltrating water can effectively advect heat and warm up the rockglacier body at depth as observed in the case of Ritigraben, which in our modelling removed the phase lag when using the observed temperature field that includes the talik formation in summer. For the other three rockglacier, water infiltration may also occur but it does not significantly warm the temperatures at depth, as confirmed by the good agreement between observed and modelled temperatures with depth, and we can therefore exclude this heat advection process. Secondly, with increasing water infiltration the water content and pore water pressure within the rockglacier material is expected to increase which in turn may reduce the shear strength and thereby enhance deformation and flow. This process has been suggested in other studies (Ikeda et al., 2008; Wirz et al., 2016b; Kenner et al., 2017; Buchli et al., 2018) and has also been proposed to explain the short-term velocity peaks with times scales of days that are related to a sudden input of water at the surface for example during the snow-melt period (Wirz et al., 2016b; Kenner et al., 2017). To what depth such water infiltration occurs is poorly known, but this process would be most effective within the shear horizon, as deformation is highest. Further, in several boreholes pressurised water was observed when drilling into the shear horizon (Arenson et al. (2002) and personal communication Alex Blast (November 2015) for Murtel rockglacier, and Buchli et al. (2018) for the Furggwanghorn rockglacier).

Regarding the multi-annual variations, that are well documented and synchronous for many rockglaciers in Switzerland (PERMOS, 2016), our modelling suggests that the responsible process between the observed acceleration in flow (e.g. from 2011 to 2015) and the observed surface warming can not be explained by heat conduction into the ground alone. It is likely an indirect effect of enhanced melt water penetrating into the rockglacier body and thereby affecting its rheology. Phases



of slowdown related to conductive cooling in cold or long winters (e.g. 2007 and 2011) are more distinct in our modelled velocities and thus winter cooling may contribute more substantially to the longer-term slow down of rockglacier flow (Fig. 4). The enhanced sensitivity to winter temperatures is (in contrast to summer temperatures) not surprising given that the zero-curtain effect basically caps the summer temperature peak at zero degrees and inhibits the propagation of the summer heat into the ground, which is well reflected in the observed temperatures below the active layer (Fig. 4).

Including a shear horizon with a pseudo-plastic rheology (with the same temperature dependency as for the main rockglacier body and enforcing the same mean flow speed) does not improve our results. In the contrary, inter-annual and seasonal variations are even more underestimated, because at the shear horizon depth, where the main deformation occurs, the signal of seasonal temperature variations from the surface is too small.

In summary, both the strong underestimation in amplitude of seasonal and multi-annual variations as well as the overestimation in time-lag of seasonal peaks in our modelling suggest that heat conductive processes can not explain the observed variation in flow speed suggesting the need for other processes such as the interaction of rockglacier rheology with surface water advecting into the rockglacier body.

## 5.3 Sensitivity experiments

The sensitivity experiments were used to explore the influence of different input geometries and parameters on the simulated surface velocity in a systematic way. The experiments are in their setup and results an extention of the earlier modelling study by Kääb et al. (2007) but here we use a more realistic model set up and rheology, and explore a more extensive parameter space.

Absolute mean velocities are strongly affected by variations in geometry, due to a changing driving stress (Eq. (6)). However, over the time scales considered in this study (seasonal to multi-annual), the geometry of a rockglacier is not expected to change substantially. For the other parameters, mean velocities are most sensitive to the bottom temperature of the rockglacier, which is somewhat representative for the thermal state of the entire rockglacier body. Again, the general thermal state of a rockglacier should not change over the considered timescales. Nevertheless, a considerable warming of a rockglacier would lead to faster flow, as also reflected in the observational datasets presented in Kääb et al. (2007). The insensitivity of the mean velocity to thermal diffusivity reflects the fact that the average thermal state of the rockglacier is not affected by uncertainties in this parameter. For high thermal conductivity values, that would require high water content and hence degrading permafrost conditions, relative seasonal variation of $16\%$ are modelled, but this remains an order of magnitude below observed seasonal velocity variations.

The model used shows some dependency on the volumetric ice content value (Fig. 5 and Fig. 6). The relative seasonal variations stay below $12\%$, even for a reduction to $40\%$ of the ice content value. For high values of ice content, the velocity peak considerably shifts in time, with a delay of 2 months ($\pi/3$ of one year cycle) in comparison to the reference scenario. Note, that the ice content of the rockglacier material is rather static and not expected to change even over time-scales of several decades.



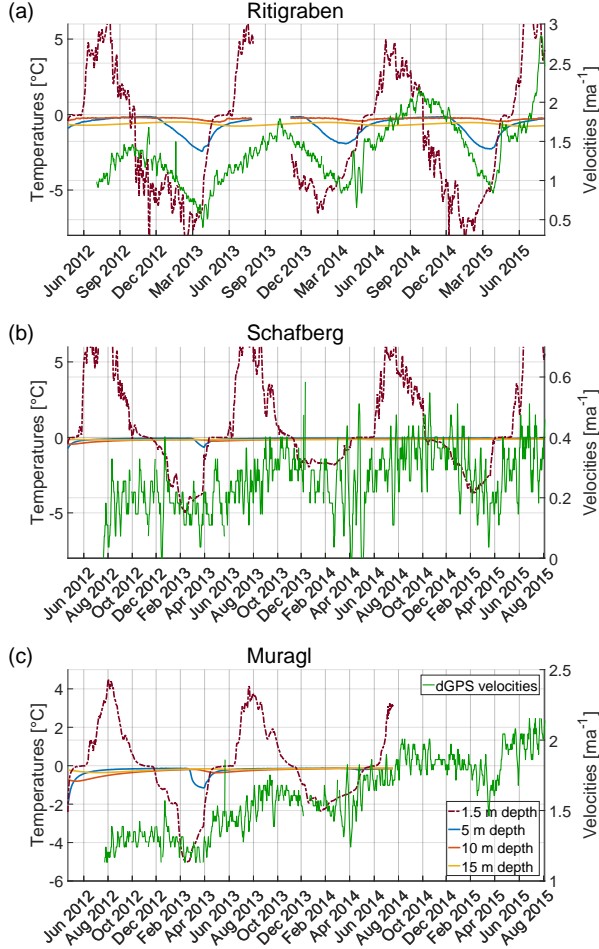

**Figure 7.** Temperature and velocity time series in the (a) Ritigraben, (b) Schafberg and (c) Muragl rockglacier. The left y-axis shows the temperatures (borehole measurements if available, model results otherwise) at $1.5\,m$ (within the active layer), $5\,m$, $10\,m$ and $15\,m$ depth. The right y-axis with the green line shows the surface velocities from *D-GPS* measurements.

Decreasing thickness values lead to very different absolute mean velocities, but more interestingly, they also lead to stronger seasonal variations. Considering realistic thickness's values of 12, 20, and 28 m, we obtain seasonal velocity variations of 21.2%, 2% and 1.6% respectively. We explain this sensitivity by the variable fraction of the rockglacier thickness that is affected by temperature variations. It implies that thin rockglaciers are more sensitive to the effect of heat conduction both for seasonal as well as for long-term temperature changes. This means that for thin rockglaciers (which are usually fast moving) heat conduction should be be considered in the interpretation of short-term variations. Note further, that without any borehole deformation data or detailed geophysical investigations, uncertainties in rockglacier thickness (see example of Schafberg) may significantly affect modelled velocities (absolute and seasonal variations).





For all other remaining parameters, except the rockglacier thickness, the modelled seasonal velocity variations do not change much and stay again below $8$ to $12\%$ of their mean flow and phase shifts vary below 2 month, even for extreme and relatively unrealistic end-member parameter values.

By considering the pseudo-plastic relation, the seasonal variations are coherently decreasing for all the scenarios (even for shallow rockglaciers) and the velocity peaks are considerably shifted in timing, with a delay up to 6 months (see Table 3). Thus

in summary, we conclude from our sensitivity study that our modelling results for the four rockglaciers above are, apart from thickness, insensitive to uncertainties in our input parameters, and the modelled magnitudes of seasonal variations and related conclusions are robust.

## 6  Conclusions

We quantitatively investigated the contribution of heat conduction to seasonal and multi-annual variations in rockglacier flow

velocity on the basis of numerical modelling and a multi-year time series of observed surface velocities and borehole temperatures from four different real-world rockglaciers. The numerical model couples heat conduction to an empirically derived rheology of rockglacier creep that accounts for temperature and ice-content. We find that using standard parameters from the literature, our modelling reproduces the right order of magnitude of mean surface velocities for all chosen rockglaciers. In contrast, the magnitudes of seasonal and multi annual variations are strongly underestimated by our modelling and the phase-lags

of the seasonal peaks too long. This suggests that the effect of heat conduction on the observed variations in surface flow is very limited and can not explain more than about $25\%$ of the observed variations. The exception are extremely thin rockglaciers, as shown in the sensitivity study (see Sect. 4.3), where short-term temperature variations can force heat conduction to affect the whole deforming thickness of the rockglacier, thus leading to more substantial velocity variations.

Additional sensitivity experiments underpin the robustness of these conclusions within expected parameter uncertainties, also when including a shear horizon at the bottom of the rockglacier. Our idealised sensitivity experiments further indicate that when the temperature changes over the full depth of the rockglacier (changing bottom temperature), the mean movement maybe affected substantially, but this requires changes in climate over periods of several decades or centuries.

From our quantitative process modelling approach we can therefore exclude heat conduction as the governing process for seasonal to multi-annual variations in rockglacier flow. Considering the phase-lag information of the summer peak (e.g. the case of case of Ritigraben) and indications from earlier qualitative and statistical analysis of rockglacier velocities (Wirz et al., 2016b), we conclude that advection of surface water into the rockglacier and its interaction over porewater pressure with the creep rheology is required to explain short-term velocity variations of rockglacier flow. However, further investigations are

required for a better understanding of the advection of water within the rockglacier material as well as of the role of water and water pressure on creep rheology.



*Data availability.* Data on rockglacier kinematics and temperature are available from the PERMOS office upon request. The reference web link is *http://www.permos.ch/data.html*.

*Competing interests.* The authors declare that they have no conflict of interest.

*Acknowledgements.* The authors want to thank all colleagues for their constructive feedback and discussions to this research. In particular we want to acknowledge the work of Johann Müller for the coherent fieldwork in Engadine, and Vanessa Wirz for her preliminary study on this topic. This work would not have been possible without the support of the PERMOS network and the PermaSense project. This study was possible due to the financing of the X-sense 2 project funded by Nano-Tera.ch (ref. 530659) and the University of Zurich.





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
