# Peer review of "Resolving the influence of temperature forcing through heat conduction on rockglacier dynamics: a numerical modelling approach"

_The Cryosphere, 2018_

## Referee Comment (RC1) · Hoelzle (Referee) · 2 Dec 2018

This manuscript models and discuss rockglacier flow and related seasonal and inter-annual variabilities by using a 1-D numerical flow model including thermal heat conduction. The model is forced by given temperature data below the active layer. Temperature data gaps were linearly interpolated. The main result shows that the overall flow velocities on an annual resolution is well simulated. However, seasonal and inter-annual variations are strongly underestimated. The authors conclude then that the heat conduction alone is not able to explain the variations and therefore non-conductive processes such as the presence of water must strongly influence the flow behavior.

General comments: This study is elaborated very carefully using a simple flow model and driving the model with available input data from the existing PERMOS network. I appreciated very much the very careful and critical discussion of their results. In general, the paper is written very concise and clear. I have only some minor points, where I suggest some small changes and additions in the paper. I would particularly enjoy when the authors would include some additional past literature and discuss them in relation to their obtained results being aware that the modelled time domain does not overlap with some past studies. However, for the discussion part this would may help to improve some interpretations.

Specific comments:

Page 1, Line 15: Please add here relevant reference: Barsch and Hell (1975). This paper is in my opinion more important than other papers mentioned here as it is directly related to one of the investigated sites of the authors.

Page 2, Line 18-19: Some earlier studies already investigated these effects, particularly on rockglacier Murtèl. (e.g. Hanson and Hoelzle 2004, Schneider et al. 2012, Scherler et al. 2014). Particularly, the paper of Schneider et al. shows clearly also that the thermal response within the active layer is very fast and is decoupled from pure heat conduction. This is especially true for the cooling process.

Page 3. Figure 1c: The borehole is according to my knowledge not at the correct place on this image. Please correct. see also paper by Hoelzle et al. 1998 describing the measurements in the boreholes at Schafberg.

Page 4, Table 1: Please explain in more detail what 'bottom temperature' mean. For example at Murtèl the total borehole thickness is about 58 m and you probably mean the temperature at 27 m depth exactly at the shear horizon. It would be nice, if the values in this table are referenced exactly with the corresponding literature.

Page 4: Line 10: Please add information also from the following reference: Herz et al.

(2003).

Page 4, line 14: here also the first model approaches by Wagner (1992) could be mentioned.

Page 5, line 1: Here many further and older relevant studies should be noted such as Kääb et al. (1998), Kääb (2002), Kääb et al. (2003).

Page 5, line 3: better to cite here the relevant papers in this context such as Arenson et al. (2010), Hilbich et al. (2009).

Page 5, line 8: The analysis of the mentioned time period was done by Arenson et al. 2002 and not in Haeberli et al. 1998.

Page 5: line 14: please have a look also at some older studies, which give some details about the old deformation measurements at the Schafberg site in relation to photogrammetric analysis (Hoelzle et al. 1998).

Page 5, line 23: Please add: Vonder Mühll and Schmid (1993).

Page 6, line 7: this is only partly true as in some papers is shown that also this rock-glacier is highly inhomogeneous such as reported in Arenson et al. (2010) and ground water is influencing the thermal regime in a depth of about 58 m: Vonder Mühll (1992)

Page 6, line 27: here maybe some more sophisticated gap filling could be used such as proposed by Staub et al. 2017.

Page 7, Figure 2b: why are at the beginning of this time period no differences plotted. data seems to be available?

Page 8, line 17: how you know the slope of ice surface without using the existing geophysical measurements?

Page 8, line 23: this is maybe not true at all sites e.g. at the borehole at Schafberg. We were never sure if the borehole was really fixed in the lower part.

Page 8, line 23: use spatial instead of special

Page 9, Figure 3 and line 5: please give full references instead of just mentioning 'proposed by literature'

Page 16, line 2: text is missing? You mean results are discussed?

Page 16, line 3: Temperature modelling -> delete s

Page 16, line 5: you could use approaches of gap filling according to Staub et 2017

Page 16, line 18 and page 17 line 1: I am not convinced if this explanation is reasonable. When we know that rockglacier Murtèl is probably the coldest of all rockglaciers, I would first assume that water may play a less important role at Murtèl than at the other rockglacier which are warmer.

Page 20, line 36: However, at Schafberg all this information (deformation, geophysics) was already published. Please refer to this literature mentioned already above.

References: ARENSON, L., HAUCK, C., HILBICH, C., SEWARD, L., YAMAMOTO, Y. & SPRINGMAN, S. (2010). Sub-surface heteorogenities in the Murtèl-Corvatsch rock glacier, Switzerland. 6th Canadian Permafrost Conference. (pp. 1494-1500). Calgary, Canada: CNC-IPA/NRC. ARENSON, L., HAUCK, C., HILBICH, C., SEWARD, L., YAMAMOTO, Y. & SPRINGMAN, S. (2010). Sub-surface heteorogenities in the Murtèl-Corvatsch rock glacier, Switzerland. 6th Canadian Permafrost Conference. (pp. 1494-1500). Calgary, Canada: CNC-IPA/NRC. BARSCH, D, Hell, G. (1975). Photogrammetrische Bewegungsmessungen am Blockgletscher Murtèl I, Oberengadin, Schweizer Alpen. Zeitschrift für Gletscherkunde und Glazialgeologie 11(2):111-142 HANSON, S., HOELZLE, M. (2004). The thermal regime of the active layer at the Murtèl rock glacier based on data from 2002. Permafrost and Periglacial Processes, 15, 273-282. HERZ, T., KING, L., GUBLER, H. (2003) Microclimate within coarse debris of talus slopes in the alpine periglacial belt and its effect on permafrost. In: Phillips M, Springman S, Arenson L (eds) 8th International Conference on Permafrost, Proceedings. Swets & Zeitlinger, Lisse, Zürich, pp 383-387 HILBICH, C., MARESCOT, L., HAUCK, C., LOKE, M. H. & MÄUSBACHER, R. (2009). Applicability of Electrical Resistivity Tomography Monitoring to Coarse Blocky and Ice-rich Permafrost Landforms. Permafrost and Periglacial Processes, 20, 269-284. HOELZLE, M., WAGNER, S., KÄÄB, A. & VONDER MÜHLL, D. (1998). Surface movement and internal deformation of ice-rock mixtures within rock glaciers at Pontresina-Schafberg, Upper Engadin, Switzerland. In A. G. A. A. LEWKOWICZ, M. (Ed. 7th International Conference on Permafrost. Proceedings. (pp. 465-471). Yellowknife, Canada: Centre d'Etudes Nordiques, Université Laval. KÄÄB, A. (2002). Monitoring high-mountain terrain deformation from repeated air- and spaceborne optical data: examples using digital aerial imagery and ASTER data. ISPRS Journal of Photogrammetry & Remote Sensing, 57, 39-52. KÄÄB, A., GUDMUNDSSON, G. H. & HOELZLE, M. (1998). Surface deformation of creeping mountain permafrost. Photogrammetric investigations on Murtèl rock glacier, Swiss Alps. In A. G. LEWKOWICZ & M. ALLARD (Eds.) Seventh International Permafrost Conference. (pp. 531-537). Yellowknife, Canada: Centre d'Etudes Nordiques, Université Laval. KÄÄB, A., KAUFMANN, V., LADSTÄDTER, R. & EIKEN, T. (2003). Rock glacier dynamics: implications from high-resolution measurements of surface velocity fields. In M. PHILLIPS, S. SPRINGMAN & L. ARENSON (Eds.) 8th International Conference on Permafrost, Proceedings. (pp. 501-506). Zürich: Swets & Zeitlinger, Lisse. DOI: 10.1002/ppp.506 SCHERLER, M., SCHNEIDER, S., HOELZLE, M. & HAUCK, C. (2014). A two sided approach to estimate heat transfer processes within the active layer of rock glacier Murtèl-Corvatsch. Earth Surface Dynamics, 2, 141-154. SCHNEIDER, S., HOELZLE, M. & HAUCK, C. (2012). Influence of surface heterogeneity on observed borehole temperatures at a mountain permafrost site in the Upper Engadine, Switzerland. The Cryosphere, 6, 517-531. STAUB, B., HASLER, A., NOETZLI, J. & DELALOYE, R. (2017). Gap-filling algorithm for ground surface temperature data measured in permafrost and periglacial environments. . Permafrost and Periglacial Processes, 28, 275-285. VONDER MÜHLL, D. (1992). Evidence of intrapermafrost groundwater flow beneath an active rock glacier in the Swiss Alps.

Permafrost and Periglacial Processes, 3, 169-173. VONDER MÜHLL, D. & SCHMID, W. (1993). Geophysical and photogrammetrical investigations of rock glacier Muragl I, Engadin, Swiss Alps. In C. GUODONG (Ed. 6th International Conference on Permafrost. Proceedings. (pp. 654-659). Beijing, China: South China University Technology Press. WAGNER, S. (1992). Creep of Alpine permafrost, investigated on the Murtèl-rock glacier. Permafrost and Periglacial Processes, 3, 157-162.

---

## Referee Comment (RC2) · Arenson (Referee) · 14 Jan 2019

        2012.

[referee-annotated manuscript omitted]

---

## Author Comment (AC1) · 18 Jan 2019

This manuscript models and discuss rockglacier flow and related seasonal and interannual variabilities by using a 1-D numerical flow model including thermal heat conduction. The model is forced by given temperature data below the active layer. Temperature data gaps were linearly interpolated. The main result shows that the overall flow velocities on an annual resolution is well simulated. However, seasonal and interannual variations are strongly underestimated. The authors conclude then that the heat conduction alone is not able to explain the variations and therefore non-conductive processes such as the presence of water must strongly influence the flow behavior.

General comments: This study is elaborated very carefully using a simple flow model and driving the model with available input data from the existing PERMOS network. I appreciated very much the very careful and critical discussion of their results. In general, the paper is written very concise and clear. I have only some minor points, where I suggest some small changes and additions in the paper. I would particularly enjoy when the authors would include some additional past literature and discuss them in relation to their obtained results being aware that the modelled time domain does not overlap with some past studies. However, for the discussion part this would may help to improve some interpretations.

**General response:**

We thank the referee for his very positive comments and critical points on which we comment in detail below. In brief we addressed the following main points:

- We added and discussed some further and indeed relevant literature in order to improve the interpretations and the discussion of the results.
- We further clarified our choice and of a linear gap filling (and related uncertainties) for the temperature input data and made clearer that this choice does not affect the conclusions of our results.

**Specific comments:**

Page 1, Line 15: Please add here relevant reference: Barsch and Hell (1975). This paper is in my opinion more important than other papers mentioned here as it is directly related to one of the investigated sites of the authors.

Answer: reference added.

Page 2, Line 18-19: Some earlier studies already investigated these effects, particularly on rockglacier Murtèl. (e.g. Hanson and Hoelzle 2004, Schneider et al. 2012, Scherler et al. 2014). Particularly, the paper of Schneider et al. shows clearly also that the thermal response within the active layer is very fast and is decoupled from pure heat conduction. This is especially true for the cooling process.

Answer: references added here in the Introduction.

Page 3. Figure 1c: The borehole is according to my knowledge not at the correct place on this image. Please correct. see also paper by Hoelzle et al. 1998 describing the measurements in the boreholes at Schafberg.

Answer: the location of the borehole has been set according to the PERMOS report No. 12-15. The location is also in accordance with the mentioned paper (Hoelzle et al. 1998), where the borehole is labeled as "borehole 2". As far as we know no correction is needed. Additionally, the reference has been added throughout the text as a first publication for the borehole.

Page 4, Table 1: Please explain in more detail what 'bottom temperature' mean. For example at Murtèl the total borehole thickness is about 58 m and you probably mean the temperature at 27 m depth exactly at the shear horizon. It would be nice, if the values in this table are referenced exactly with the corresponding literature.

Answer: for more clarity the text in the caption has been corrected: "rockglacier bottom temperature, measured at the lower end of the shear horizon". In section 3 Data and Methods, this points is further explained (Page 7, line 7).

We added references in the caption of the table for the ice content (Hoelzle et al. 1998, Arenson and Springman 2002) and for the temperature and deformation data (Permos).

Page 4: Line 10: Please add information also from the following reference: Herz et al. (2003).

Answer: in the text we discuss the development of a Talik at depth. The suggested publication discuss heat advection only in the blocky active layer of Ritigraben rockglacier.

Page 4, line 14: here also the first model approaches by Wagner (1992) could be mentioned.

Answer: We would like here to cite the more relevant publications: Loewenherz et al, 1989, and Kääb et Weber, 2004.

Page 5, line 1: Here many further and older relevant studies should be noted such as Kääb et al. (1998), Kääb (2002), Kääb et al. (2003).

Answer: Kääb et al. (1998) has been added. The text has been corrected adding "amongst others" before the citations.

Page 5, line 3: better to cite here the relevant papers in this context such as Arenson et al. (2010), Hilbich et al. (2009).

Answer: references added: Arenson et al. (2010).

Page 5, line 8: The analysis of the mentioned time period was done by Arenson et al. 2002 and not in Haeberli et al. 1998.

Answer: as far as we know, the data are presented already in Haeberli et al. 1998. The reference (Arenson et al. 2002) has been added.

Page 5: line 14: please have a look also at some older studies, which give some details about the old deformation measurements at the Schafberg site in relation to photogrammetric analysis (Hoelzle et al. 1998).

Answer: reference added (Hoelzle et al. 1998). The text has been added: "Hoelzle et al. (1998) investigated internal deformation profiles from borehole measurements in relation to photogrammetric analysis".

Page 5, line 23: Please add: Vonder Mühll and Schmid (1993).

Answer: reference added. The text has been corrected: "Older geophysical and photogrammetrical measurements have been presented in Vonder Mühll (1993)".

Page 6, line 7: this is only partly true as in some papers is shown that also this rockglacier is highly inhomogeneous such as reported in Arenson et al. (2010) and ground water is influencing the thermal regime in a depth of about 58 m: Vonder Mühll (1992)

Answer: I think the comment refers to line 5. Here, to account for this comment, we corrected the text to: "[…]of the creeping rockglacier. All the parameters are assumed to be homogeneous in time and space in first approximation. Note that geophysical investigations (Arenson 2002, Arenson et al. 2010 amongst others) showed that rockglaciers can be highly heterogeneous".

Page 6, line 27: here maybe some more sophisticated gap filling could be used such as proposed by Staub et al. 2017.

Answer: We agree that more sophisticated gap filling would deliver more accurate filling of the gaps. However, we believe that for the purpose of the study this more sophisticated gap filling is not needed and would not change any of the conclusions made here. First of all, the gaps are scarce, and our focus are seasonal as well as multi-annual variations, which, as we state already in the text (p. 6 line 25-27 in original version) we interpret purposely the whole times series outside the periods of the gaps. We tried to further clarify this point and changed the text to: "This approach is considered satisfactory due to the combination of the scarcity of data gaps in the surface temperature time series and the length of the considered seasonal to multi-annual time scales. Note that more sophisticated methods (e.g. Staub et al. 2016) could be used for interpolating the time series".

Page 7, Figure 2b: why are at the beginning of this time period no differences plotted. data seems to be available?

Answer: data are plotted but not visible. The problem lies in the graphical interpolation for plotting as a contour-plot in the used software (Matlab). The measurements for the first years were manually done, i.e. only very little profiles are available until 1994 and they are not visible in the last graph.

Page 8, line 17: how you know the slope of ice surface without using the existing geophysical measurements?

Answer: the ice surface is assumed to be parallel to the surface slope. The text has been corrected: ", which is assumed to be parallel to the surface slope".

Page 8, line 23: this is maybe not true at all sites e.g. at the borehole at Schafberg. We were never sure if the borehole was really fixed in the lower part.

Answer: the text has been corrected: "This hypothesis is confirmed by the low deformation rates at the bottom of the borehole inclinometer profiles for all the studied rockglaciers.".

Page 8, line 23: use spatial instead of special

Answer: typo corrected.

Page 9, Figure 3 and line 5: please give full references instead of just mentioning 'proposed by literature'

Answer: reference added (Vonder Mühll, 1993; Arenson et al., 2002; Lugon and Stoffel, 2010).

Page 16, line 2: text is missing? You mean results are discussed?

Answer: error corrected. We indeed meant "discussed". The text has been deleted after the second referee's comment.

Page 16, line 3: Temperature modelling -> delete s

Answer: typo corrected.

Page 16, line 5: you could use approaches of gap filling according to Staub et 2017

Answer: see comment for page 6 line 27 above.

Page 16, line 18 and page 17 line 1: I am not convinced if this explanation is reasonable. When we know that rockglacier Murtèl is probably the coldest of all rockglaciers, I would first assume that water may play a less important role at Murtèl than at the other rockglacier which are warmer.

Answer: our hypothesis is confirmed by Arenson et al. 2010, being one of the main conclusions. The reference has been added and the text has been corrected: "flow" instead of "content".

Page 20, line 36: However, at Schafberg all this information (deformation, geophysics) was already published. Please refer to this literature mentioned already above.

Answer: the text has been corrected, deleting the example of Schafberg.

References: ARENSON, L., HAUCK, C., HILBICH, C., SEWARD, L., YAMAMOTO, Y. & SPRINGMAN, S. (2010). Sub-surface heterogeneities in the Murtèl-Corvatsch rock glacier, Switzerland. 6th Canadian Permafrost Conference. (pp. 1494-1500). Calgary, Canada: CNC-IPA/NRC.

ARENSON, L., HAUCK, C., HILBICH, C., SEWARD, L., YAMAMOTO, Y. & SPRINGMAN, S. (2010). Sub-surface heterogeneities in the Murtèl-Corvatsch rock glacier, Switzerland. 6th Canadian Permafrost Conference. (pp. 1494-1500). Calgary, Canada: CNC-IPA/NRC.

BARSCH, D, Hell, G. (1975). Photogrammetrische Bewegungsmessungen am Blockgletscher Murtèl I, Oberengadin, Schweizer Alpen. Zeitschrift für Gletscherkunde und Glazialgeologie 11(2):111-142

HANSON, S., HOELZLE, M. (2004). The thermal regime of the active layer at the Murtèl rock glacier based on data from 2002. Permafrost and Periglacial Processes, 15, 273-282.

HERZ, T., KING, L., GUBLER, H. (2003) Microclimate within coarse debris of talus slopes in the alpine periglacial belt and its effect on permafrost. In: Phillips M, Springman S, Arenson L (eds) 8th International Conference on Permafrost, Proceedings. Swets & Zeitlinger, Lisse, Zürich, pp 383-387

HILBICH, C., MARESCOT, L., HAUCK, C., LOKE, M. H. & MÄUSBACHER, R. (2009). Applicability of Electrical Resistivity Tomography Monitoring to Coarse Blocky and Ice-rich Permafrost Landforms. Permafrost and Periglacial Processes, 20, 269-284.

HOELZLE, M., WAGNER, S., KÄÄB, A. & VONDER MÜHLL, D. (1998). Surface movement and internal deformation of ice-rock mixtures within rock glaciers at Pontresina-Schafberg, Upper Engadin, Switzerland. In A. G. A. A. LEWKOWICZ, M. (Ed. 7th International Conference on Permafrost. Proceedings. (pp. 465-471). Yellowknife, Canada: Centre d'Etudes Nordiques, Université Laval.

KÄÄB, A. (2002). Monitoring high-mountain terrain deformation from repeated air- and spaceborne optical data: examples using digital aerial imagery and ASTER data. ISPRS Journal of Photogrammetry & Remote Sensing, 57, 39-52.

KÄÄB, A., GUDMUNDSSON, G. H. & HOELZLE, M. (1998). Surface deformation of creeping mountain permafrost. Photogrammetric investigations on Murtèl rock glacier, Swiss Alps. In A. G. LEWKOWICZ & M. ALLARD (Eds.) Seventh International Permafrost Conference. (pp. 531-537). Yellowknife, Canada: Centre d'Etudes Nordiques, Université Laval.

KÄÄB, A., KAUFMANN, V., LADSTÄDTER, R. & EIKEN, T. (2003). Rock glacier dynamics: implications from high-resolution measurements of surface velocity fields. In

M. PHILLIPS, S. SPRINGMAN & L. ARENSON (Eds.) 8[th] International Conference on Permafrost, Proceedings. (pp. 501-506). Zürich: Swets & Zeitlinger, Lisse. DOI: 10.1002/ppp.506

SCHERLER, M., SCHNEIDER, S., HOELZLE, M. & HAUCK, C. (2014). A two sided approach to estimate heat transfer processes within the active layer of rock glacier Murtèl-Corvatsch. Earth Surface Dynamics, 2, 141-154.

SCHNEIDER, S., HOELZLE, M. & HAUCK, C. (2012). Influence of surface heterogeneity on observed borehole temperatures at a mountain permafrost site in the Upper Engadine, Switzerland. The Cryosphere, 6, 517-531.

STAUB, B., HASLER, A., NOETZLI, J. & DELALOYE, R. (2017). Gap-filling algorithm for ground surface temperature data measured in permafrost and periglacial environments. . Permafrost and Periglacial Processes, 28, 275-285.

VONDER MÜHLL, D. (1992). Evidence of intrapermafrost groundwater flow beneath an active rock glacier in the Swiss Alps. Permafrost and Periglacial Processes, 3, 169-173.

VONDER MÜHLL, D. & SCHMID, W. (1993). Geophysical and photogrammetrical investigations of rock glacier Muragl I, Engadin, Swiss Alps. In C. GUODONG (Ed. 6th International Conference on Permafrost. Proceedings. (pp. 654-659). Beijing, China: South China University Technology Press.

WAGNER, S. (1992). Creep of Alpine permafrost, investigated on the Murtèl-rock glacier. Permafrost and Periglacial Processes, 3, 157-162.

First I'd like to apologize for the delay in submitting my comments. Unforeseeable circumstances did not allow me to review the paper earlier.

Cicoira et al. present a well written manuscript on a numerical study of rock glacier creep with the objective of understanding seasonal changes in surface deformation. Overall the manuscript reads well and follows a logical flow. Assumptions and limitations of the model are reasonably explained. I also do, in general, agree with the author's conclusion. One aspect I'd like to see included is a presentation of the depths of the zero annual amplitude, and a discussion that relates those depths with the limited C1 TCD Interactive comment Printer-friendly version Discussion paper success of explaining seasonal changes in deformation via conduction. With temperatures being the driver for the creep, no seasonal change in the velocity would be expected if the temperatures remain constant.

The manuscript could also benefit from some editorial improvements and I made several suggestions in the annotated version attached to this comment.

Please also note the supplement to this comment:
https://www.the-cryosphere-discuss.net/tc-2018-176/tc-2018-176-RC2- supplement.pdf

**General response:**

We thank the referee for his constructive and thorough feedback and critical points on which we comment in detail below. In brief we addressed the following main points:

- We addressed the comment about the depths of zero annual temperature amplitude (see details below).
- We implemented the editorial improvements suggested by the referee (see details below).
- We addressed all the more specific comments in the list below (some little corrections are omitted).

**Specific comments:**

Page 1, line 6:
Answer: "flow velocity" is now used consistently throughout the whole paper. Hereon, this correction will be omitted from the author review response.

Page 2, line 2:
Answer: the following sentence has been added: "The single-word term rockglacier is used here instead of "rock glacier" consistently to Barsch, 1996.".

Page 2, line 7:
Answer: the term "identify" has been changed to "suggested".

Page 2, line 9:
Answer: the text has been corrected: "Possible positive feedback processes between rising temperature and increased deformation have also been suggested (Ikeda et al., 2008; Buchli et al., 2018)".

Page 2, line 30-31:
Answer: Reference and weblinks have been added.

Page 3, line 6:
Answer: the rockglaciers. The text has been corrected accordingly.

Page 3, line 15:
Answer: reference (Arenson et al., 2004) has been added.

Page 4, line 5:
Answer: the text has been corrected adding: "relative to other rockglaciers.".

Page 4, line 7:
Answer: the text has been corrected throughout the whole paper adding: "volumetric" to ice content.

Page 6, line 1:
Answer: the text has been corrected: "…using the software MATLAB (2016).".

Page 7, Figure 2:
Answer: the units have been added to the figure, the caption has been corrected. Also the scales in the lower subpanels have been changed to better show the relative difference in temperature.

Page 7, eqn 1:
Answer: reference has been added (Williams and Smith, 1989).

Page 13, line 19:
Answer: the value has been corrected, it was indeed 18%.

Page 13, line 33:
Answer: the information about the depth of the zero annual amplitude for conduction has been added to the section Results in the paragraph about temperature modelling (page 16, line20 in the manuscript).
"The depth of the zero annual temperature amplitude for the studied rockglaciers is according to the PERMOS (2016) report usually between 10 and 20 m depth. This depth is slightly overestimated by our model (see Fig.2), with the seasonal temperature signal influencing reaching slightly deeper in comparison to the borehole measurements (PERMOS, 2016)."
The authors find a direct comparison between this depth and the depth of the shear surface to be misleading. Instead of doing so, the text of the discussion has been modified at page 18 line 17 in the manuscript.
"In consequence, comparing rockglacier thermal regime  and rockglacier surface flow velocities is non-trivial and interpretation potentially misleading".

Page 13, line 20:
Answer: the text has been corrected adding the comparison. "more sensitive than to the previous parameters".

Page 16, line 28:
Answer: the text has been corrected: "For Murtèl, the mean surface velocities (averaged over the whole time series) match the observations.".

Page 17, line 28:
Answer: the text has been corrected. The new sentence is "The influence of interstitial water on creep is partially implicitly taken into account in the adopted empirical creep relation through the dependency on

temperature; but note, the impact on the stress regime due to pore pressure is in the adopted rheology not taken into account.".

Page 19, Line 5:
Answer: a reference to the PERMOS report 2016 has been added along the value of the temperature variations observed in the studied boreholes. The text has been implemented: "too small, being close to one tenth of a degree (PERMOS, 2016).".

Page 19, Line 25:
Answer: The text has been clarified here: "The model used shows a dependency of the surface velocities on the volumetric ice content value.

Page 21, Line 10:
Answer: The text has been deleted, because it was a repetition of what already included in the conclusion.

Page 21, Line 29:
Answer: The text has been implemented: "short- to mid-term".

We thank the referees and the editor for the constructive review. On top of the answers to the referees we want to edit some minor details in the manuscript. The specific changes are commented below.

**Specific comments:**

Page 3, line 1: The section title "Case Study Sites" has been changed in "Study Sites".

Page 22, line 7: The acknowledgements section has been slightly changed to include the D-GPS pilot project of BAFU, the editor and the referees.

[revised manuscript text omitted]